# Development and Performance Comparison of a 64-Channel Annular Array Probe Excited Simultaneously by Shorted Symmetrically Positioned Elements

**DOI:** 10.3390/s25041221

**Published:** 2025-02-17

**Authors:** Shintaro Fukumoto, Takahiro Arakawa

**Affiliations:** 1IHI Inspection and Measurement Co., Ltd., 2-6-17 Fukuura, Kanazawa-ku, Yokohama 236-0004, Japan; 2Certification for Inspection of Welds (CIW) Inspection Industry Association, General Incorporated Association, Chairman, Former IHI Inspection and Measurement Co., Ltd., 5-4-5 Asakusa Bashi, Taito-ku, Tokyo 111-0053, Japan; arakawa115@jcom.home.ne.jp

**Keywords:** non-destructive testing, ultrasonic phased array, point focusing, annular array, sector scan, phased array

## Abstract

Ultrasonic testing technology is used to inspect pipe welds in nuclear and thermal power plants. This paper proposes a new method to measure weld defects in thick-walled pipes of about 100 mm using ultrasonic phased array technology. The effectiveness of annular arrays is confirmed by numerical simulations, and element arrangements that enable point focusing and sector scanning are considered. The energy concentration of annular arrays is 7% higher than that of linear arrays and 3% higher than that of matrix arrays. Similarly, the sound pressure ratio of grating lobes is equivalent to that of linear arrays and 20% lower than that of matrix arrays. This array probe is driven by 64 channels by dividing the ring of an 8-element annular array probe in parallel and shorting the elements at symmetrical positions. The effectiveness is examined by measuring specimens with flat-bottom holes and simulated spherical defects. The authors confirmed peaks in the echo intensity of a φ1 mm flat-bottom hole and a φ3 mm pseudo-spherical defect arranged at 5 mm intervals. Comparing the measured results with a conventional linear array transducer, the results from the proposed method show that the number and size of defects can be accurately measured.

## 1. Introduction

UT utilizes the received waveforms that are generated when ultrasonic waves propagate through a material and are reflected, mode converted, and diffracted by defects, etc., and is based on simple geometric relationships, assuming that the speed of sound in the material is constant [1,2]. Measurements are generally performed using a single-element probe and imaging the results of the flaw detection requires time and effort. In contrast, phased array is a technology that uses an array of tens to hundreds of regularly arranged elements to control and transmit signals, and then phase-matches the received waves at each element to synthesize a high-resolution flaw detection image. With the help of the excitation function, which means the delay time of the individual transducers, the sound field can be focused and swiveled to a required angle depending on the calculated delay time [3,4,5,6]. Furthermore, the focusing behavior of ultrasonic beams has been confirmed [7]. Around 2000, linear array probes were the mainstream approach among phased arrays [8,9,10]. A 32-channel linear array has been used for the inspection of titanium round alloy bars during production [11], and more recently matrix array probes have been used, with a 126-channel matrix array being used for quality control of steel pipe welds in steel mills [12]. Robots are used to automate the production line where steel pipes are produced. When using such a multi-channel phased array, it is suitable for line and robot automation, but the large size of the probe and measuring device makes it difficult to apply it to manual flaw detection. In addition, while more elements improve focusing and steering performance and allow for a wider inspection range, the cost of both the probe and the equipment is also increased. Advances in computer technology have allowed the miniaturization and generalization of phased-array flaw detection equipment, increasing its suitability for practical applications. In Japan, ultrasonic testing is generally performed to inspect the welds of steel pipes for defects. Standards call for a single-probe ultrasonic test [13]. Furthermore, flaw detection conditions should be optimized to evaluate the state of dense microvoids that occur in the initial state of creep damage in extremely thick pipes [14,15,16,17].

There are two typical phased array ultrasonic scanning methods. The first is the linear scanning method, which scans the ultrasonic beam parallel or perpendicular to the inspection surface, and the second is the sector scanning method, which scans the ultrasonic beam in a fan shape. In either case, the position and direction of the ultrasonic beam can be easily changed, and it is expected that the position and dimensional accuracy of defects can be improved [18]. Point-focusing flaw detection improves the accuracy of flaw size measurement [19] and was introduced as an excellent method in the round robin test results of PISC II, an international joint research project involving 15 countries worldwide [20]. This method is suitable for evaluating minute flaws, such as creep damage.

However, point-focusing flaw detection is rarely used in phased-array flaw detection, and line-focusing flaw detection using a linear array probe is more common [21]. Conventional linear array probes can utilize these two scanning methods; however, focusing the ultrasonic beam in the width direction of each element is not possible. On the other hand, conventional annular array sensors can focus on a point and can perform vertical linear scanning but cannot perform sector scanning. Generally, linear array inspection is more widely used [22,23]. This is due to the element arrangement, and the probe used in this paper has an element arrangement that improves on these.

In order to carry out quality inspections of the inside of welds after on-site welding for thermal power boilers in Japan, it is basically necessary to follow domestic standards. Articles 79 and 80 of the Enforcement Regulations of the Electricity Business Act [24] contain provisions on the scope of voluntary welding inspections [25], and the non-destructive testing method for voluntary welding inspections is specified as radiographic testing (RT) by the Interpretation of the Technical Standards for Thermal Power Plant Equipment for Power Generation [26,27], and no other means are specified. However, the inspection method for areas not subject to voluntary welding inspections based on the Enforcement Regulations of the Electricity Business Act is not specified as RT. However, when RT is carried out on-site, it is necessary to set up a radiation control area, and welding and repair work other than inspection cannot be carried out at the same time. For this reason, UT is required as an alternative to RT [28]. One example of an important criterion for performing UT is the ASME Code [29]. In recent years, Ni-based alloys have been adopted in A-USC boilers, and austenitic welds exist. Austenitic welds, such as Ni-based alloys and stainless steel, have a lower signal-to-noise (S/N) ratio than general steel materials because the ultrasonic beam is scattered and attenuated by the effect of columnar crystals [30]. High-resolution detection of small, deep defects is required because traditional linear arrays may not be able to detect them.

In this paper, the authors design element arrangements that can detect with higher accuracy than conventional linear array probes for thick-walled pipes of about 100 mm to which UT is applied, and measure test pieces with flat-bottom holes and simulated spherical defects to investigate its effectiveness. Section 2 describes the design of the ultrasonic probe. The degree of focusing of the ultrasonic beam is defined by an equation called the energy concentration degree and is compared for each phased array. Section 3 describes the manufacture of the ultrasonic probe and the results of inspection using test pieces with flat-bottom holes and simulated spherical defects. Section 4 provides conclusions.

## 2. Transducer Design

### 2.1. Ultrasonic Energy Concentration

A common method for measuring the focusing characteristics of a focusing probe is by determining the beam width by scanning a horizontal hole back and forth and obtaining the beam width from the probe movement range where an echo height of 1/2 or more of the maximum echo height can be obtained. However, for a line-focusing probe, the beam width in the plate thickness direction and in the horizontal direction differ substantially, making it difficult to compare the flaw detection ability with that of a point-focusing probe. In addition, the directivity may be diminished markedly in the range of 1/2 or less of the maximum echo height, making it difficult to use as an objective indicator.

The authors defined the ratio of ultrasonic energy received by a certain microscopic area as the ultrasonic energy concentration of the transmitted ultrasonic energy. The characteristics of the focusing probe were measured using this evaluation method to discuss the detectability of minute flaws.

At the focal point of the ultrasound beam, the sound pressure distribution in a 100 × 100 mm *xy* plane perpendicular to the direction of travel of the ultrasound beam was determined at 1 mm intervals. For the sound pressure at each position, *p_i_*, and maximum sound pressure, *p_max_*, the ultrasonic energy concentration, *n*, in 1 mm^2^ was defined as the ratio. Additionally, with the sound pressure of the focused ultrasonic beam set to 0 dB and the device positioned at the center of a 100 × 100 mm *xy* plane, it was confirmed that the sound pressure remains saturated at −12 dB or less within the 100 × 100 mm *xy* plane. Moreover, in terms of the relationship between area and energy concentration, the concentration level is almost constant for a focused probe of 100 × 100 mm or more. However, a larger size was chosen, considering that the same method applies to cases where the probe does not converge.*n* = (*p_max_*)^2^/Σ(*p_i_*)^2^(1)

### 2.2. Transducer Dimensions

Evaluating creep damage of steel pipe joints at nuclear and thermal power plants requires ultrasonic flaw detection to a depth of approximately 75 mm. Using the ultrasonic energy concentration, *n*, the dimensions of the transducer required to focus the energy of the ultrasonic beam to a depth of 75 mm were investigated with ultrasonic flaw detection numerical simulation software (CIVA2023 SP1, EXTENDE, Paris, France), which is internationally trusted and used in many studies [31,32]. The nominal frequency was 5 MHz, the nominal refraction angle was 45°, the method was shear wave oblique angle flaw detection, and the circular vibrators were 10, 20, 35, and 50 mm in diameter.

Figure 1 shows the ultrasonic energy concentration of each transducer dimension in an annular array probe that is concentrically divided into 32 parts with uniform width. This result was obtained by using ultrasonic numerical simulation to obtain the sound pressure distribution of each probe, substituting it into Equation (1), and calculating the energy concentration. For comparison, the results for a single oscillator of 10 × 10 mm without focusing are also shown. For a nominal refraction angle of 45°, the beam path length was approximately 105 mm at a depth of 75 mm. As the beam path length increased, the ultrasonic energy concentration decreased, and a larger transducer diameter was required to increase the ultrasonic energy concentration at a deeper position. However, there was no significant difference in the ultrasonic energy concentration around the beam path length of 105 mm between diameters of 35 and 50 mm. An ultrasonic energy concentration of nearly 9% was obtained at a depth of around 75 mm (beam path length of around 105 mm). This concentration was approximately twice the ultrasonic energy concentration for a 20 mm diameter transducer, and the selection of the transducer size could improve flaw detection accuracy. A practical and efficient transducer size of 35 mm in diameter was chosen, after which the number of element divisions was considered.

### 2.3. Number of Resonator Divisions

A 35 mm diameter transducer divided into 32 parts resulted in an ultrasonic energy concentration of 9%. The effect of the number of divisions was investigated. For comparison, an annular array probe, a linear array probe, and a matrix array probe were also examined.

In the linear array probe, a single transducer with a length of 35 mm (width 20 mm) was divided linearly into elements of the same size to eliminate the effect of ultrasound energy concentration due to the difference in the number of elements. In the annular array probe, a transducer with a diameter of 35 mm was divided concentrically so that the rings had the same width. The diameter of the center circle with the smallest diameter was made equal to the width of each ring. The matrix array probe had piezoelectric elements of 35 × 35 mm so that the element size and the number of vertical and horizontal divisions were the same. The pitch width between each element was 0.05 mm.

Each probe was attached to a shear wave wedge (density 1.18, longitudinal sound velocity 2680 m/s) with a wedge distance of 35 mm and a nominal refraction angle of 45°, and the focus was set at a depth of 75 mm. The ultrasonic energy concentration was determined.

For the linear array and annular array probes, the energy concentration increased with the number of elements, and when the probes were divided into eight or more elements, the energy concentration approached that obtained when the probes were divided into 32 elements (Figure 2). Point focusing using an annular array probe achieved 3.5 times higher energy concentration than line focusing using a linear array probe. In matrix array probes, increasing the number of elements should increase the energy concentration further, but the same energy concentration as an annular array probe would require 100 elements (vertically and horizontally). Approximately 10 divisions were required for both.

### 2.4. Effect of Grating Lobes

The difference in the generation of grating lobes between the annular array probe and the matrix array probe was also measured. Figure 3 shows an example of a linear sound pressure distribution centered on the position of the maximum sound pressure and the peak of the grating lobe is visible. The sound pressure distribution was obtained for a 64-channel matrix array probe in which a 35 × 35 mm transducer was divided vertically and horizontally into 8 sections, and strong grating lobes were generated. The ratio between the maximum sound pressure of this main pole and the maximum value of the grating lobes on both sides was measured as the sound pressure ratio of the grating lobes (Figure 4). The annular array probe had the smallest grating lobe sound pressure ratio, whereas the matrix array probe had the largest. For example, for the matrix array probe, the grating lobe sound pressure ratio was approximately 13% even when it was divided into 100 elements (10 vertically and horizontally), whereas the annular array probe with only 4 elements had a much smaller grating lobe sound pressure ratio.

### 2.5. Beam Width

Figure 5 shows the beam width obtained from the sound pressure distribution. Because the sound pressure distribution was for the transmitted waves, the beam width that provided a sound pressure 3 dB lower than the maximum echo height was determined and plotted. That is, the beam width was measured at a sound pressure that was 6 dB lower (1/2) than the maximum sound pressure at the echo height from a small horizontal hole, which is a function of the product of the directivity of the transmitted wave and the directivity of the received wave. 

The annular array probe narrowed the ultrasonic beam width to near the minimum beam width, even with the smallest number of elements, but the matrix array probe and the linear array probe affected the quantitative evaluations such that flaw detectability could not be ascertained.

The ultrasonic energy concentration indicated that point focusing using an annular array probe could concentrate ultrasonic energy with a smaller number of elements and reduce the generation of grating lobes. The results suggested that an element size of about 35 mm in diameter would give an ultrasonic energy concentration of about 9% at a depth of about 75 mm.

### 2.6. Element Division Method

Figure 6 illustrates an element division method devised to achieve sector scanning and excellent point focusing characteristics with an annular array probe. The array probe was designed with 113 elements by dividing the rings of an 8-element annular array probe in parallel. The number of elements was 113, but by short-circuiting the elements at symmetrical positions, the pulses excited the elements at symmetrical positions with the same delay law, creating an array probe driven by 64 channels. In addition, the element pitch must satisfy the following condition to prevent grating lobes from occurring. Grating lobes are a phenomenon in which ultrasonic waves are focused in positions other than the focusing direction. This phenomenon is determined by the element pitch *d* (mm), wavelength *λ* (mm), and refraction angle θ (°). The refraction angle was set to 45°, which follows Snell’s law and prevents longitudinal wave components from entering during angle beam inspection [33].(2)$$d\geq \frac{\lambda }{1+\sin\theta }$$

In research aimed at improving phased array UT technology, a matrix array probe using 80 elements has been developed [34,35]. As in this study, the performance of three-dimensional scanning of the ultrasonic beam was evaluated by sound field analysis, and the arrangement of the transducers was determined. However, the probe designed in this section can be used with 64 channels while using 113 elements by shorting the symmetrical elements on the left and right. This reduces the number of channels used, leading to cost savings and system miniaturization.

## 3. Results

### 3.1. Verification of Sector Scanning Performance Through a Numerical Simulation

The sector scanning performance of the probe described in Section 2.6 was verified by using a numerical simulation. For comparison, numerical simulation results were used for a linear array probe in which the element was divided into 15 parts with the same ele-ment dimensions. In the linear array probe, a transducer with a length of 35 mm and a width of 20 mm was divided into 15 elements of the same shape by parallel straight lines, and the frequency was 5 MHz. In a numerical simulation, it was difficult to divide the el-ements as shown in Figure 6, so the elements were divided approximately by dividing each ring radially as shown in Figure 7. The centers of the element widths coincided.

Figure 8 shows the simulation results. The beam profile that enabled sector scanning with the sector scanning annular array probe was approximately the same as that of a linear array probe. Equivalent sector scanning was possible when the ring-shaped vibrator was linearly divided and when the rectangular vibrator was linearly divided, provided that the number of divided elements was equal. In addition, in sector scanning with re-fraction angles of 37.4° to 56.3°, multiple focal points were set at equal intervals from a depth of 60 to 85 mm in a straight line with a 45° inclination to the flaw detection surface. A sharp focusing range was observed for the ultrasonic beam at all refraction angles, suggesting that accurate flaw detection would be possible.

### 3.2. Experimental Testing of Flaw Detection Performance with Artificial Flaws

#### 3.2.1. Test Overview

The probe was fabricated according to the specifications shown in Figure 6, and its ability to detect flaws was confirmed. An optical photograph of the fabricated probe is shown in Figure 9. The probe in this optical photograph is equipped with a wedge with an oblique angle of 45 degrees. The probe used a composite piezoelectric element with a diameter of 34.4 mm, the element spacing was 0.1 mm, the wedge was made of acrylic, and the distance within the wedge was 35 mm. The specifications and acoustic characteristic tolerance levels of this probe and the linear array probe used for comparison are shown in Table 1. Figure 10 shows the dimensions of each probe. The ultrasonic flaw detector was MultiX (M2M, France). Flaws were detected using sector scanning on a wedge with a nominal refraction angle of 45°, using two specimens with flat-bottomed holes and spherical flaws at a depth of around 75 mm.

#### 3.2.2. Verification with Flat-Bottomed Holes

Flat-bottomed holes of various sizes were created centered at a depth of 75 mm, and the detectability of flaws was examined. The flat-bottomed holes had diameters of 200 μm to 2.0 mm (Figure 11). Flat-bottomed holes with a length of 10 mm were introduced using a flat-bottomed drill, but because the length of the flat-bottomed hole with a diameter of 200 μm was 4 mm, which was the machining limit, the center depth was 79.2 mm. Figure 12 shows an optical photograph of the test piece.

The flaw detection conditions were sector scanning with refraction angles of 33.7° to 56.3°, with the focus set at a depth of 90 mm when the refraction angle was 33.7°, and at a depth of 60 mm when the refraction angle was 56.3°; the focus position changed linearly at the corners. For comparison, a 64-channel linear array probe with element dimensions of 0.9 × 20 mm (0.1 mm pitch) was used. The measurement resolution is 1 mm for both electronic and manual scanning.

The probe position was set to detect a flaw with a depth of 75 mm at a refraction angle of 45°, and the probe was scanned linearly from side-to-side while performing sector scanning. The C-scan results are shown in Figure 13. The horizontal axis of the flaw detection image represents the probe position, and the vertical axis represents the maximum echo height in the scan order of the sector scan. The echo height at the bottom is 80% of the full scale, and the flaw detection image is amplified further by 12 dB. 

With the linear array probe, adjacent flaws at intervals of 5 to 10 mm in the longitudinal direction could not be separated and appeared as continuous long flaws. The indications of adjacent flaws near the center, which was detected at a refraction angle of 45°, were connected, and the echo height resulted from interference between the echoes of multiple flaws and could not be evaluated as the echo height of a single flaw. Therefore, the linear array probe may greatly overestimate the length.

With the sector scanning annular array probe, all flat-bottomed holes could be separated, and the range of flaws could be correlated with the size of the flat-bottomed holes, allowing highly accurate evaluation. Furthermore, the probe detected flat-bottomed holes with a diameter of 200 μm, which could not be detected with the linear array probe, with an S/N ratio of approximately 5 dB, but it was recognized that this would be difficult to confirm from this result. Figure 14 shows the results of ultrasonic reflection waves (A-scan) from a 200 μm diameter flat-bottom hole. The reflection waves obtained from a 200 μm diameter flat-bottom hole in the sector scanning ring array probe are larger than the noise level, but smaller than the noise level in the linear array.

The echo height of the sector scanning annular array probe was 3.5 to 6.7 dB higher than that of the linear array probe for flaws away from the center. The noise level around the flaw was −43.4 to −46.4 dB for the sector scanning annular array probe and −44.0 to −45.9 dB for the linear array probe, which were similar. Because the plane on which the ultrasonic beam was perpendicularly incident was used as the comparison flaw, the noise level was also similar, and the difference in echo height was the difference in the S/N ratio.

#### 3.2.3. Verification with Spherical Flaws

A test specimen containing spherical flaws with a diameter of 3 mm arranged in a crisscross pattern at 5 mm intervals was prepared by diffusion bonding. Before bonding, hemispheres with a diameter of 3 mm were fabricated on the two bonding surfaces, and then the positions of the two hemispheres were aligned to create the spherical flaws when the surfaces were joined. Figure 15 shows an outline of the test specimen. Figure 16 shows an optical photograph of the measurement conditions for this test piece. For sensitivity adjustment, a 20 × 20 mm planar flaw was fabricated. The flaw detection method was the same as in Section 3.2.2, but when the sensitivity was adjusted, the echo height from the planar flaw was set to 80% of the full scale, and the flaw detection image was amplified by 27 dB.

Figure 17 shows the sector scanning annular array probe clearly separated and imaged flaws with a higher echo level than the linear array probe. Furthermore, similar to the flatbottomed holes, the linear array probe could not separate adjacent flaws, and the echoes interfered.

## 4. Discussion and Conclusions

This paper has described a phased array element for the high-resolution detection of several millimeters of false defects at a depth of about 75 mm for the evaluation of creep damage in steel tubular joints in nuclear and thermal power plants. By adding parallel line division to the concentric circle division of the annular array probe, an element arrangement capable of sector scanning and point focusing was obtained through numerical simulation. For the phased array probe, a circular oscillator with a diameter of 35 mm was divided concentrically into 8 elements and 15 linear rows, and the elements at symmetrical positions were short-circuited to create a 64-channel system excited with the same delay law. The maximum ratio of ultrasonic energy received per 1 mm^2^ was defined as the ultrasonic energy concentration. In the simulation, the energy concentration was 3.5 times greater at a depth of 75 mm than with a linear array probe with the same aperture width. As for the energy concentration, since the sector scanning annular array probe has 8 rings, the linear array has 32 channels, and the matrix array has 64 channels. As a result, it was confirmed that the energy concentration of the sector scanning annular array probe is 7% higher than that of the linear array and 3% higher than that of the matrix array. Similarly, it was confirmed that the sound pressure ratio of the grating lobe of the sector scanning annular array probe is equivalent to that of the linear array and 20% lower than that of the matrix array. The numerical simulation used in this paper uses the Rayleigh integral based on the Kirchhoff–Helmholtz integral equation [36] to calculate ultrasonic radiation, and an approximation using ray bundles for diffusion. This method can accurately approximate the main polarization component of the radiation wave, but the approximation of the cross-polarization component is not very accurate. Applying the finite element method to the numerical simulation and calculating the sound field from the driving surface of the transducer is considered more effective. A sector scanning annular array probe was created with this element arrangement, and its effectiveness was confirmed by measuring a test piece with a flat-bottom hole and a simulated spherical defect. Figure 18 shows the ultrasonic peak rate in the electronic scanning direction of the φ1 mm flat-bottom hole in Figure 13. Both the sector scanning annular array and the linear array were able to confirm five flat-bottom holes separately. The sector scanning annular array was able to confirm a higher peak rate, with a maximum difference of 60%. Figure 19 shows the ultrasonic peak rate in the manual scanning direction of the φ1 mm flat-bottom hole in Figure 13. The sector scanning annular array was able to confirm five flat-bottom holes separately. On the other hand, the linear array was not able to separate them. This is because the focused ultrasonic beam width is larger in the linear array.

In future work, the suitability of the ultrasonic energy concentration as an index for evaluating flaw detectability will be examined. Furthermore, it is anticipated that this method can be applied not only to the inspection of steel pipe welds but also to measuring the depth and length of cracks in bolts. The target cracks are fatigue cracks, and because they are closed cracks, high resolution is required for detection. Additionally, when probing from the head of the bolt, these cracks may occur at a depth of about 100 mm. The linear array can detect the depth of the crack [37]; however, it cannot detect the length. Therefore, it is expected that the sector scanning annular array can detect both the depth and the length of the crack.

## Figures and Tables

**Figure 1 sensors-25-01221-f001:**
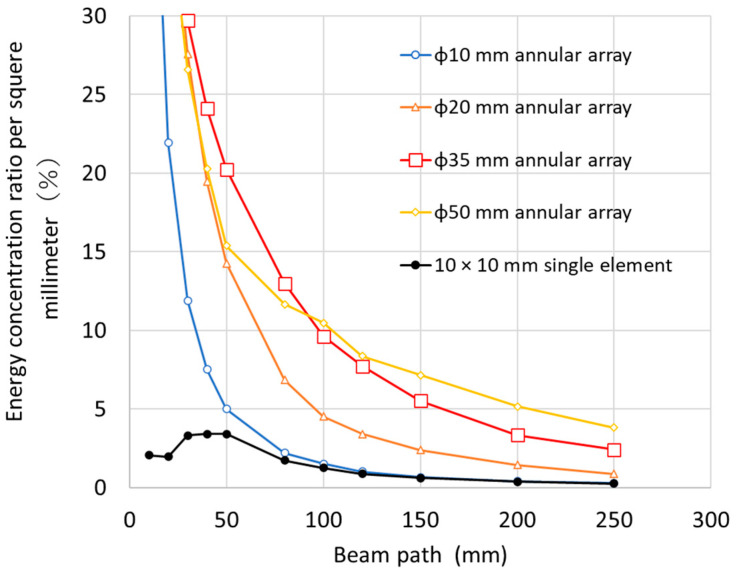
Relationship between the energy concentration ratio per square millimeter and beam path showing the effect of the aperture size of the annular array probe.

**Figure 2 sensors-25-01221-f002:**
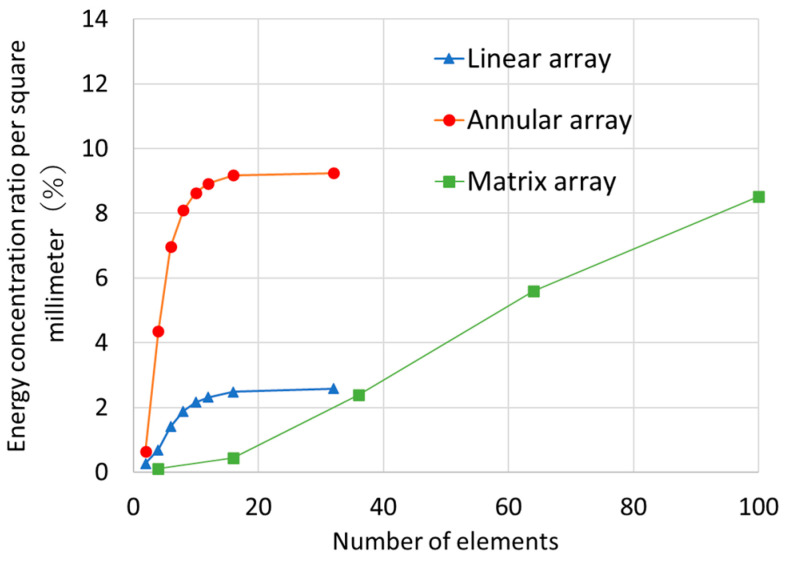
Effect on the energy concentration ratio per square millimeter of the number of uniformly sized elements a 35 mm transducer is divided into for each type of phased array probe.

**Figure 3 sensors-25-01221-f003:**
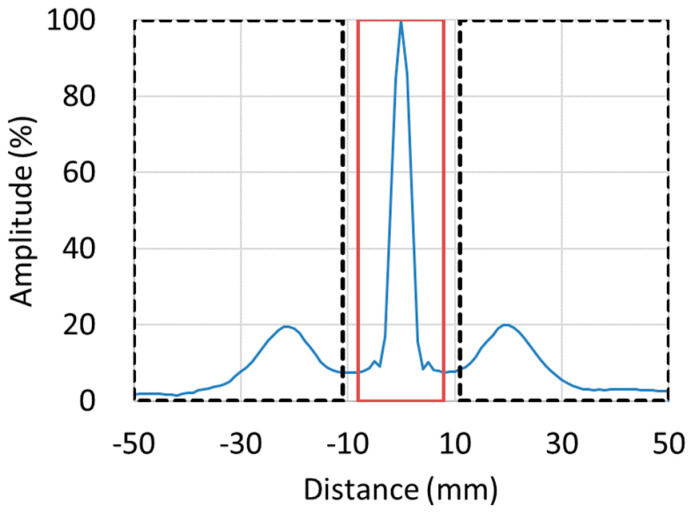
Definition of the grating lobe sound pressure ratio (example shown for matrix array probes with 64 channels).

**Figure 4 sensors-25-01221-f004:**
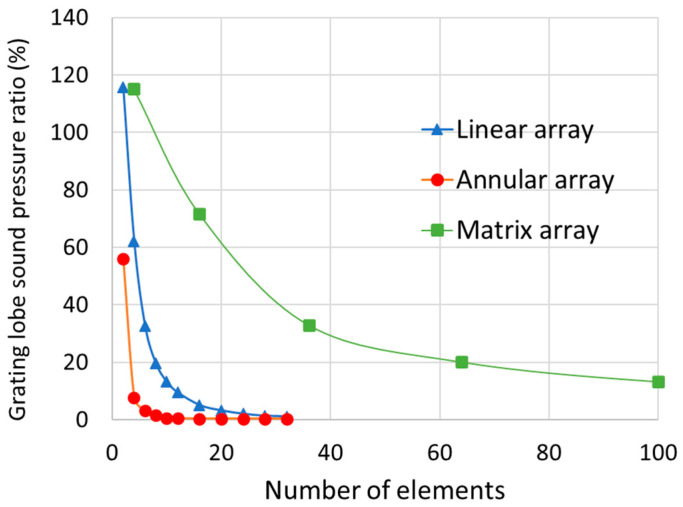
Effect of the number of elements of equal width into which a 35 mm transducer is divided on the sound pressure ratio of the grating lobe for each type of phased array probe.

**Figure 5 sensors-25-01221-f005:**
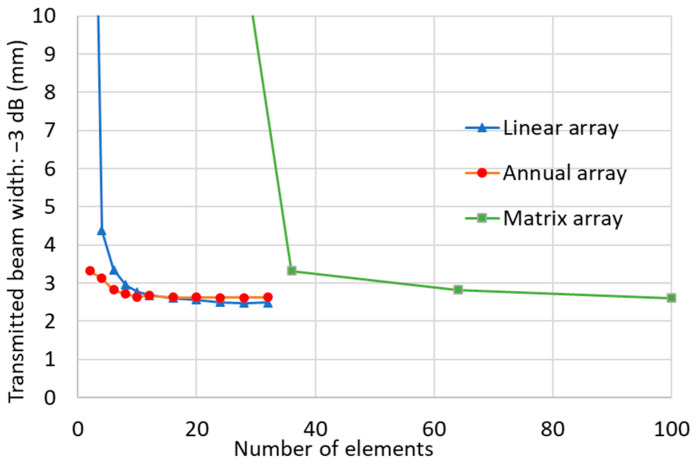
Effect of the number of elements of equal width into which a transducer of 35 mm was divided on the transmitted beam width measured by the 3 dB reduction method for each type of phased array probe.

**Figure 6 sensors-25-01221-f006:**
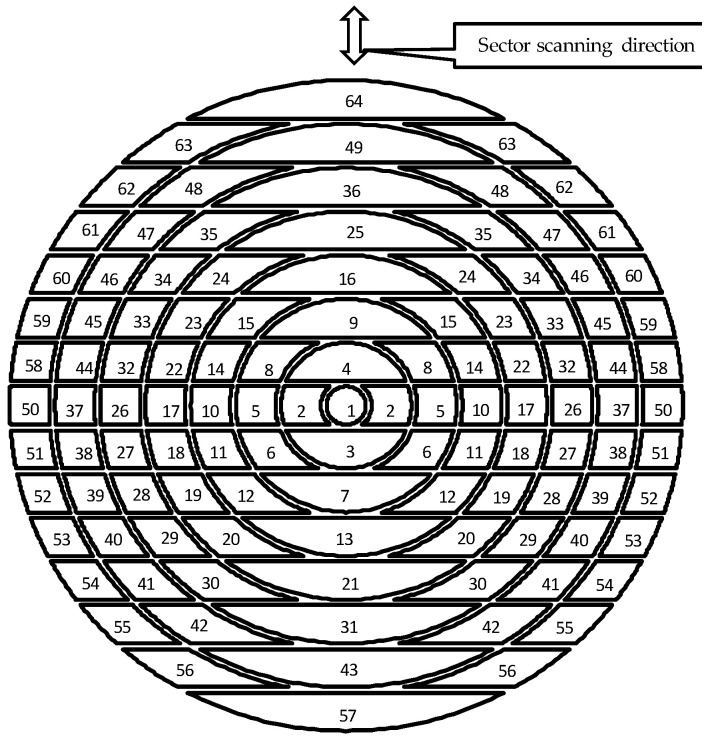
Division of the transducer in the sector scanning annular array probe.

**Figure 7 sensors-25-01221-f007:**
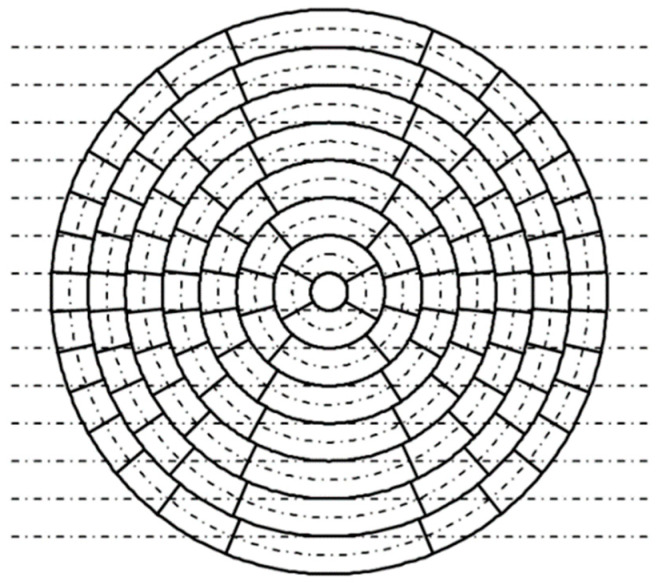
Division method for transducer used for an ultrasonic numerical simulation.

**Figure 8 sensors-25-01221-f008:**
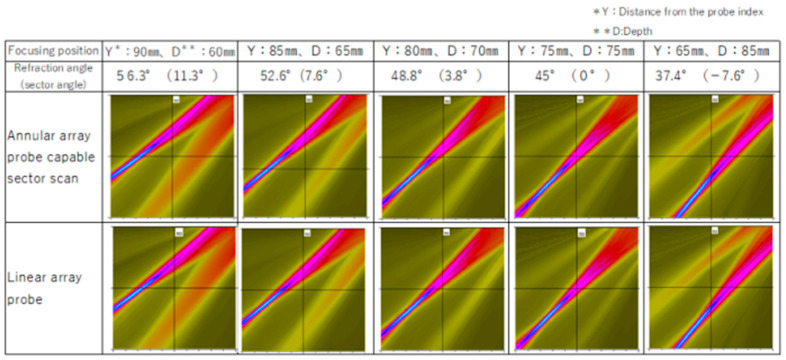
Comparison of sector scanning capability of the annular array probe and linear array probe.

**Figure 9 sensors-25-01221-f009:**
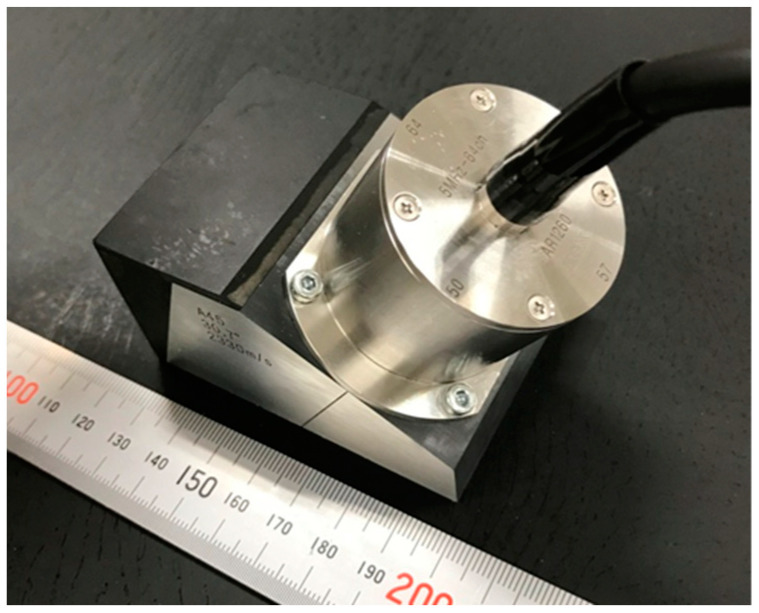
An optical photograph of the sector scanning annular array probe.

**Figure 10 sensors-25-01221-f010:**
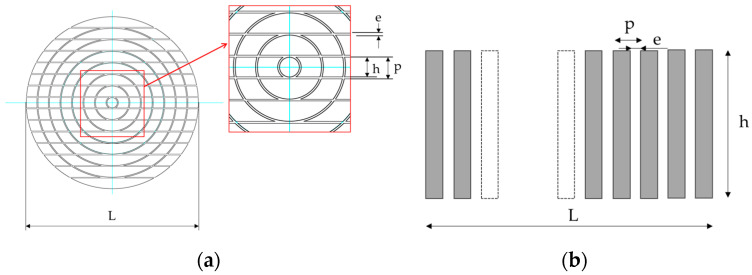
The dimensions with the (**a**) sector scanning annular array probe and (**b**) linear array probe.

**Figure 11 sensors-25-01221-f011:**
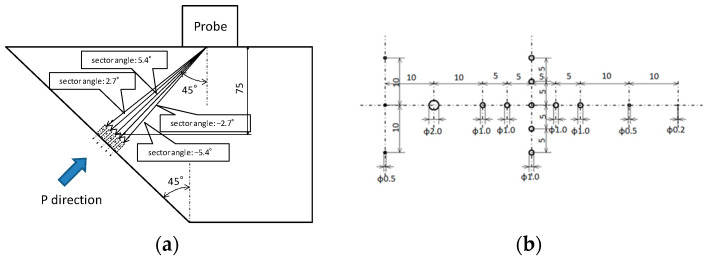
Configuration of the test piece with flat-bottomed holes of several sizes with the (**a**) lateral direction and (**b**) P-direction view.

**Figure 12 sensors-25-01221-f012:**
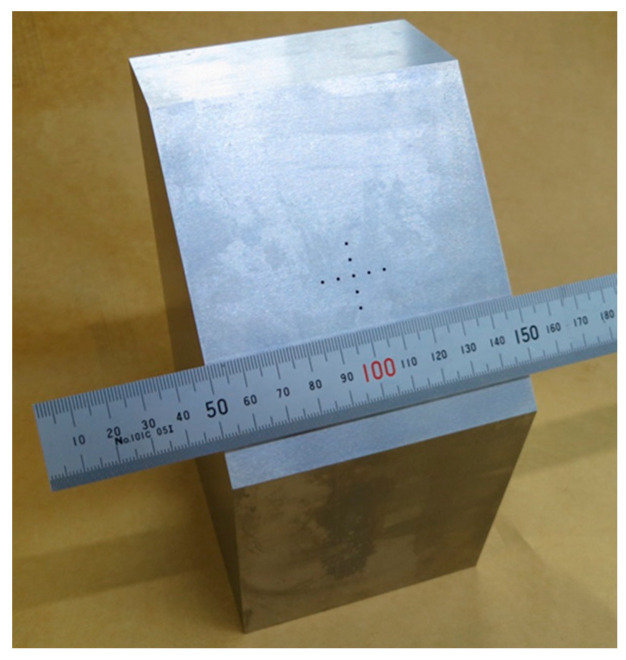
An optical photograph of the test piece with flat-bottomed holes of several sizes.

**Figure 13 sensors-25-01221-f013:**
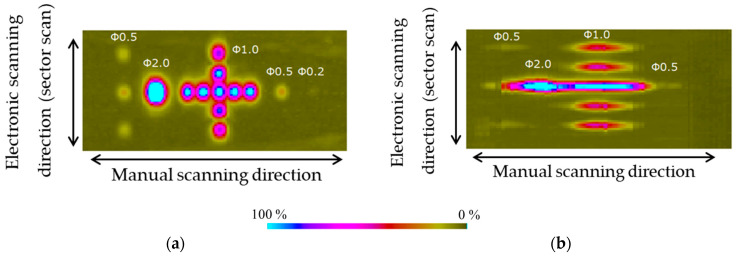
Phased array images (C-scan) of flat-bottomed holes of several sizes using sector scanning obtained with the (**a**) sector scanning annular array probe and (**b**) linear array probe.

**Figure 14 sensors-25-01221-f014:**
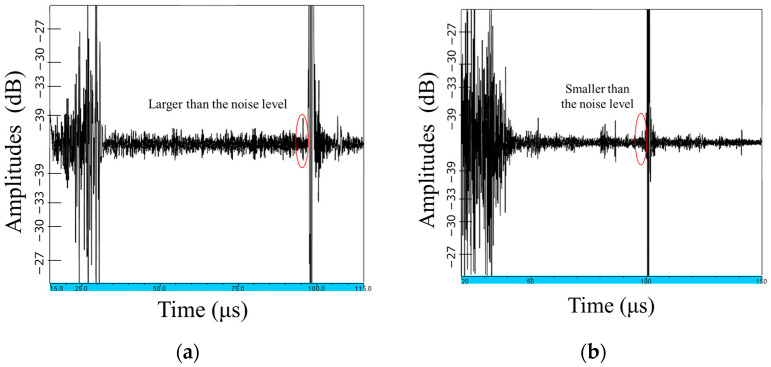
Ultrasonic reflection waves (A-scan) from a 200 μm diameter flat-bottom hole with the (**a**) sector scanning annular array probe and (**b**) linear array probe.

**Figure 15 sensors-25-01221-f015:**
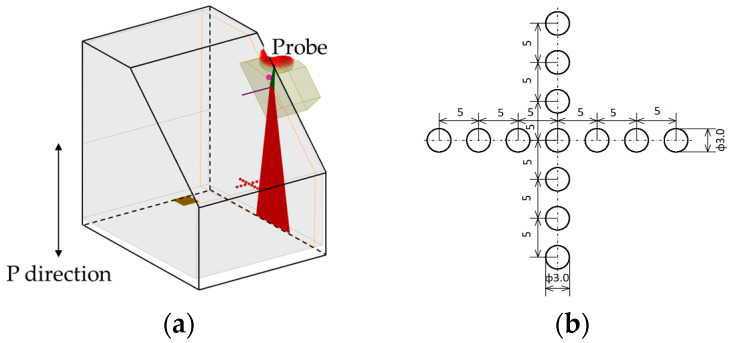
Configuration of test piece with embedded spherical flaws 3 mm in diameter with the (**a**) lateral direction and (**b**) P-direction view.

**Figure 16 sensors-25-01221-f016:**
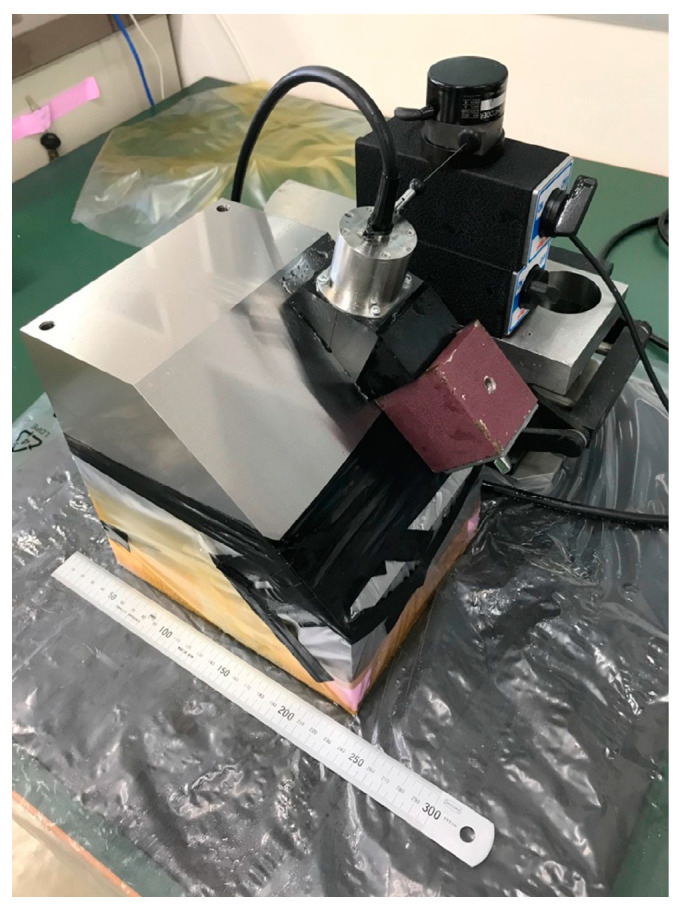
An optical photograph of the diffusion bonded spherical defect specimen.

**Figure 17 sensors-25-01221-f017:**
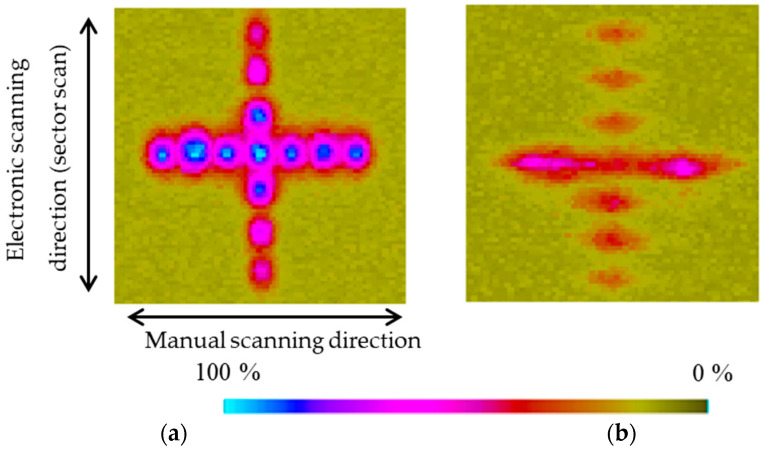
Phased array images of embedded spherical flaws 3 mm in diameter obtained by sector scanning with the (**a**) sector scanning annular array probe and (**b**) linear array probe.

**Figure 18 sensors-25-01221-f018:**
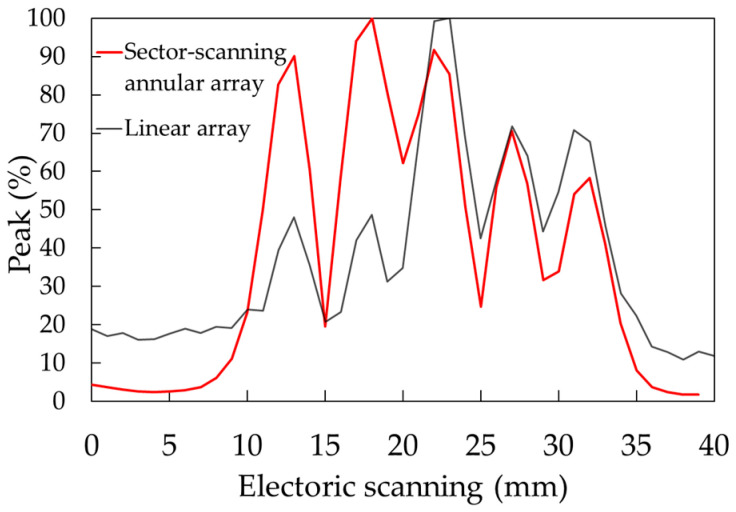
The ultrasonic peak rate in the electronic scanning direction of the φ1 mm flat-bottom hole.

**Figure 19 sensors-25-01221-f019:**
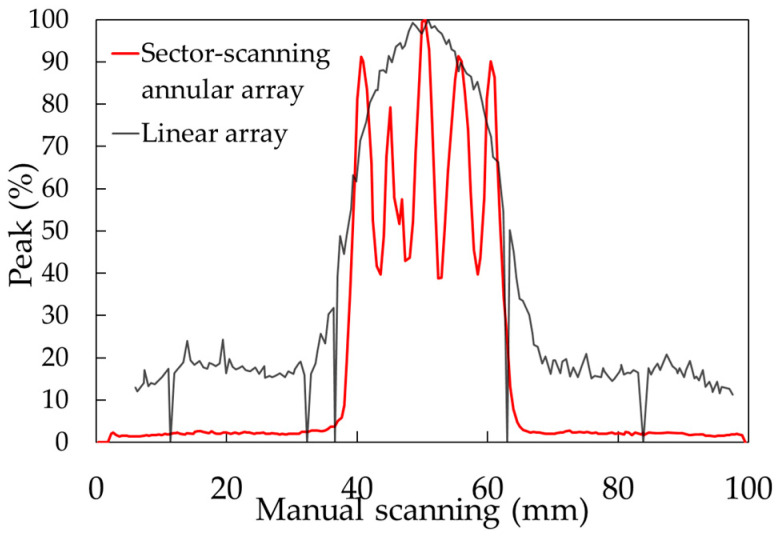
The ultrasonic peak rate in the manual scanning direction of the φ1 mm flat-bottom hole.

**Table 1 sensors-25-01221-t001:** The specifications and acoustic characteristic tolerance levels of the sector scanning annular array probe and the linear array probe.

Array Type	Number of Channels	Number of Elements	Elementary Pitch (p)	Inter Element Spacing (e)	1ch Element Outer Diameter (h)	Width of the Element (h)	Total Active Length (L)	Centre Frequency
Annular array probe excited simultaneously by shorted symmetrically positioned elements	64	8 rings × 15 rows = 113 elements	2.3 mm	0.1 mm	φ2.2 mm	-	φ34.4 mm	5 MHz
Linear array	64	64	1.0 mm	0.1 mm	-	20 mm	35.0 mm	5 MHz

## Data Availability

Data are contained within the article.

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
