# Peer review of "Development and Performance Comparison of a 64-Channel Annular Array Probe Excited Simultaneously by Shorted Symmetrically Positioned Elements"

_sensors, 2025, doi:10.3390/s25041221_

Round 1

Reviewer 1 Report

Comments and Suggestions for Authors

This is a very interesting work,providing an effective design of a sector-scanning 64-channel annular array probe.  However,  why cannot conventional linear arrays be sector-scanned? More explanation and references are needed in the introduction section.

Besides, how to optimize the design of 64-element annular array probe need to provide a more detailed theoretical approach and formula derivation for readers.

Optical photograms of the array probe and test piece are missing.

The detection of the 200um-diameter flat-bottomed holes is not very credible, please provide more detailed experimental data.

In Figs.10 and 12, the necessary numerical description is missing in the color bar.

In summary, this manuscript is suggested to be overhauled before publication.

Reviewer 2 Report

Comments and Suggestions for Authors

In this work, a sector-scanning annular array probe was developed. Numerical simulation calculations showed that this annular array probe a had higher ultrasonic energy concentration ratio and lower grating lobe generation than the linear array probe and matrix array probe with the same transducer aperture size.

The main comments of the manuscript are as follows:

 In line 85, please explain the principle for determining the sound pressure distribution is 100 × 100 mm.

Reviewer 3 Report

Comments and Suggestions for Authors

Dear Authors,

the manuscript entitled " Development and performance comparison of a 64-channel annular array probe excited simultaneously by shorted symmetrically positioned elements" by Shintaro Fukumoto and Takahiro Arakawa deals with development of probe for ultrasonic testing of flaws in pipes used in steel pipe joints at nuclear and thermal power plants. The Authors design transducer and conducted the simulation tests. After that the verification of the probe was performed. The topic of the article is very interesting and the practical application is relevant to the safe operation of nuclear power plants. I appreciate the contribution that the Authors made in design, simulation and experimental testing of ultrasonic probe as well as preparing the manuscript. However, in my opinion the manuscript needs to be improved in some fields and some general remarks as well as the specific comments are bellow.

Evaluation of the paper, general remarks

-  The Abstract section should present quantitative results and not only the most important qualitative results and/or generic considerations. The authors have written: “Numerical simulation calculations showed that this annular array probe a had higher ultrasonic energy concentration ratio and lower grating lobe generation than the linear array probe and matrix array probe with the same transducer aperture size”. What was the percentage difference? Significant improvements are expected in this section of the manuscript.

- Line 38 – the authors have written: “Phased array transducers often have 16 to 128 elements but can have as many as 256[6-10]”. This sentence refer to 5 articles at once. It makes sense to describe in detail these articles if they are important to the research presented by the Authors in the manuscript. Please add details of this articles in Introduction section.

- What is the scientific goal of the research? The description presented in the article at the end of chapter 1 looks like a functional objective and not the scientific goal: In this paper, we developed an annular array probe capable of 64-channel sector scanning by adding linear divisions to the concentric element divisions, and we compared the annular array probe with linear and matrix phased array probes using a numerical simulation.”. The next lines in this paragraph discuss the characteristics of the proposed measurement system. In my opinion, this should be included in the passages on the methodology of the design and development of the measurement system. What is the main novelty of the article? What does the article contribute to the current state of knowledge? What scientific gap does it fill? This information is missing.

- The main goal of the article is to design an array probe capable of 64-channel sector scanning by adding linear divisions to the concentric element divisions. The Introduction section lacks information on other such systems used in UT measurements. Nor is the methodology for developing similar solutions for steel pipe verification described. In addition, the references include articles from the 1980s, and only 2 articles out of 23 are from the last 5 years. Therefore, the Introduction chapter needs a complete rewrite and the addition of current references from the article's subject matter.

- Figure 1 - Are the results shown in this figure the results of simulation research or probe testing?

- Table 1 - the table is formatted not in line with Sensors journal requirements. Please read the instructions for authors – journal template. Why were such characteristics of the head (system) chosen for the measurements? If the probe is to be used in steel pipe joints at nuclear and thermal power plants, please indicate the specific standards and the resulting requirements and relate them to the assumptions that the measurement system must meet.

- Please add the view of the probe and measuring system during the evaluation of samples with wholes.

- Authors present their results but without any discussion supported by the literature. When the results are not discussed and conveniently supported by the open literature, questionable conclusions are obtained. Currently, the article looks more like a report from simulation and test than a scientific article. Significant modification are required.

The above modifications should be implemented before considering the manuscript for publication. I hope these suggestions can help to improve the quality of this paper.

I wish you all the best.

Comments on the Quality of English Language

Editorial comments/typos:

- Line 12 – there is: “…we developed…”, the Authors use the personal form. This is not correct in high-quality articles. It suggests modifying this part of the article. The same remark for line: 40, 61, 62, 81 and others. Please check entire manuscript and provide the modifications.

- Line 19 – there is: “…probe a had higher…” and should be probe had higher.

Round 2

Reviewer 1 Report

Comments and Suggestions for Authors

The quality of the manuscript has been further improved after the revision and now it can be accepted.

Reviewer 3 Report

Comments and Suggestions for Authors

Dear Authors,

The article has been significantly revised. I thank the authors for the corrections made, which were suggested in the first review. Nevertheless, some corrections are still necessary.

- the authors did not remove the personal form from the article. In several places there are still: "we", "they". Please check the entire manuscript and modify those parts of the text that are written in the personal form.

- in the final version of the manuscript, please remove the crossed-out sentences.

I wish you all the best

Reviewer
